# Managing Colorectal Cancer from Ethology to Interdisciplinary Treatment: The Gains and Challenges of Modern Medicine

**DOI:** 10.3390/ijms25042032

**Published:** 2024-02-07

**Authors:** Monika Berbecka, Maciej Berbecki, Anna Maria Gliwa, Monika Szewc, Robert Sitarz

**Affiliations:** 1Department of Human Anatomy, Medical University of Lublin, 20-950 Lublin, Poland; monikapilecka88@gmail.com (M.B.); anna.maria.gliwa@gmail.com (A.M.G.);; 2General Surgery Ward, Independent Health Center in Kraśnik, 23-200 Kraśnik, Poland; mberbecki@gmail.com; 3I Department of Surgical Oncology, Center of Oncology of the Lublin Region, St. Jana z Dukli, 20-090 Lublin, Poland

**Keywords:** colorectal cancer, epidemiology, screening, prophylaxis, risk factors, biomarkers, prevention, cyclooxygenase, genes polymorphism

## Abstract

Colorectal cancer (CRC) is a common malignant tumor of the gastrointestinal tract, which has become a serious threat to human health worldwide. This article exhaustively reviews colorectal cancer’s incidence and relevance, carcinogenesis molecular pathways, up-to-date treatment opportunities, prophylaxis, and screening program achievements, with attention paid to its regional variations and changes over time. This paper provides a concise overview of known CRC risk factors, including familial, hereditary, and environmental lifestyle-related risk factors. The authors take a closer look into CRC’s molecular genetic pathways and the role of specific enzymes involved in carcinogenesis. Moreover, the role of the general practitioner and multidisciplinary approach in CRC treatment is summarized and highlighted based on recent recommendations and experience. This article gives a clear understanding and review of the gains and challenges of modern medicine towards CRC. The authors believe that understanding the current patterns of CRC and its revolution is imperative to the prospects of reducing its burden through cancer prevention and cancer-adjusted treatment.

## 1. Introduction

The recent World Health Organization (WHO) evaluations show a significant increase in the incidence of malignant tumors within the aging population [1,2]. Up to 2035, deaths from colorectal cancer (CRC) and rectal cancer are predicted to increase. In Latin American and Caribbean countries, the number of deaths is predicted to double by 2035 [3]. Cancer is the second-leading cause of death in the world and is one of the most significant health and economic issues for society [1]. CRC carcinogenesis—a transformation from adenoma to cancer—is a long-lasting process [4]. On the one hand, it simplifies successful screening in risk groups; on the other hand, early diagnostics at the time of initial, non-characteristic symptoms may lead to the detection of this disease in a fully curable stage. The morbidity and mortality associated with CRC can be reduced by implementing preventive measures such as targeted screening programs and early therapeutic intervention. Adapting one’s lifestyle, following an appropriate diet, regular physical activity, and the maintenance of weight all play a key role in minimizing the risk of CRC [5].

This document updates and systematizes the available management strategies for CRC.

## 2. The State of Knowledge from Symptoms to Novel Molecular Biology Achievements

### 2.1. Incidence of Colorectal Cancer and Its Geographic Variations

Throughout the world, CRC is nowadays a major cause of cancer-related morbidity and mortality, with nearly 1.9 million new cases diagnosed and almost 935,000 deaths in 2020 [1].

The global distribution of CRC has large geographic differences, with the number of new cases rapidly increasing due to population growth, changes in demographics, and the Westernization of lifestyle habits [6].

The highest incidence of CRC occurs in Europe, Australia/New Zealand, and North America [1]. CRC is both the third most commonly diagnosed cancer and the third most common cause of cancer-related deaths in both men and women in the United States [5]. In 2020, CRC was the most common type of cancer incidence among men in Slovakia, the countries of the Arabian Peninsula, Ethiopia, and Southeast Asian countries such as Singapore and Brunei Darussalam. It was also reported that CRC is the cancer with the highest mortality among men in countries of the Arabian Peninsula and Ethiopia and among women in Spain, Croatia, Belarus, Estonia, and Japan [1]. Two-thirds of CRC cases occur in countries characterized by high or very high indices of development and/or income. A pathological examination in approximately 95% of CRC cases reveals adenocarcinomas and other types of cancers that occur, including mucinous carcinomas and adenosquamous carcinomas [1,6].

The overall incidence of CRC increased in most European countries in recent decades. The annual increase is calculated in different countries and lies between 0.4% and 3.6%. Statistics also show racial and ethnic differences in CRC survival. There was a 20% higher incidence rate and approximately 40% higher mortality rate among non-Hispanic Blacks or African Americans of all racial and ethnic groups in the USA. Data from the National Centers for Disease Control and Prevention show that 1 in 41 African American males die from CRC compared to 1 in 48 for whites, and for African American women, 1 in 44 die from CRC compared to 1 in 53 white females [7]. Research has confirmed that African Americans have a higher risk of developing CRC and a higher risk of dying from the disease [7,8]. Asian American and Pacific Islander patients have an 8% to 10% lower cancer-related mortality rate compared with non-Hispanic White patients. Among the subgroups of Asian Americans and Pacific Islanders (Chinese, Japanese, and Filipino), Chinese men had the lowest cancer-specific mortality [8].

Interestingly, it was observed that, in North America, Australia, and China, the incidence of CRC among the population between 20 and 40 years increased; in Europe, this was observed among subjects aged 20–49 years. This implies the need to evaluate screening programs; The American Cancer Society recommended lowering the age to start screening from 50 to 45 years [9].

### 2.2. Temporal Trends in Colorectal Cancer Incidence, Mortality and Survival Rates

The incidence of CRC worldwide is constantly increasing, especially in countries characterized by a high-income economy and high human development index (HDI) [5,6,10]. CRC is considered one of the clearest markers of the cancer transition, replacing infection-related cancers in countries undergoing rapid social and economic changes. Such changes in the causes of cancer are predominantly linked to Western lifestyles and are, thus, frequently found in high-income countries. Given the temporal profiles and demographic projection, it is expected that by 2030, the global number of CRC cases will increase by 60% to more than 2.2 million new cases and approximately 1.1 million deaths annually [6]. Nowadays, rapid increases in both CRC incidence and mortality are observed in many medium-to-high HDI countries in Eastern Europe, Asia, and South America. In several of the highest indexed HDI countries (the USA, Australia, New Zealand, and Western Europe countries), CRC incidence and mortality rates have shown to stabilize or even decline, which may be a long-term result of rising social awareness regarding this disease, early detection and prevention through polypectomy [11].

Although over the past few decades, effective therapeutic strategies for CRC have been developed, the five-year overall survival is unsatisfactory. This is caused by the presence of the following poor prognostic factors: vascular, neural invasion, a low lymphocyte-to-monocyte ratio (LMR), late diagnosis, and tumor stage [12]. It is estimated that approximately 20% of CRC patients have already progressed into a metastatic state at the time of presentation, and more than 30% of patients with early CRC eventually develop metastatic disease [13,14]. CRC treatment’s long-term costs remain a huge social and economic burden [15]. Hence, a huge emphasis is placed on personalized medicine (PM), which may be scientifically translated to tailoring medical treatment: correlations between a patient’s personal history, genomic profile, and specific biomarkers seem to be responsible for responses to medications and cancer genetic variants [6,16].

### 2.3. Symptoms of Colorectal Cancer

The course of CRC can be asymptomatic for a long time; the appearance of clinical symptoms is often indicated with an advanced stage of cancer. CRC symptomatology depends on the tumor localization (Table 1). The classic symptoms of CRC include explicit or latent bleeding from the gastrointestinal tract (rectal bleeding or blood in the stool), abdominal pain, a palpably perceptible tumor, the rhythm of bowel movement disturbance (alternating constipation and diarrhea), gastrointestinal tract obstruction, unintended weight loss, fatigue, anemia, and febrile states [17,18]. Explicit bleeding from the lower digestive tract is common (in 50–60% of patients with CRC), and an easily observable symptom (per rectum examination) requires the determination of its cause in each case [19,20]. Another very common symptom is a bowel movement disorder (constipation/diarrhea), which occurs in about 50% of CRC cases; if it appears for more than 6 weeks without apparent cause, this becomes an alarming symptom [21]. An important symptom for CRC patients, which has been stated by various authors at a frequency of 30% to more than 75%, is anemia [18,22]. When diagnosing men over 40 years of age, as well as post-menopausal women with iron deficiency, such as anemia, the exclusion of a proliferative process is required. Weight loss, a palpable tumor, and abdominal pain are significantly associated with high levels of colorectal cancer.

The key to a successful early CRC diagnosis is performing a thorough physical examination. Moreover, the patient’s diet, medication, medical, surgical, and psychological history can prove to be valuable information during the medical examination, with some being closely correlated with CRC (e.g., anemia, hematochezia, unintentional weight loss, or a family history of CRC or inflammatory bowel disease—IBS). A differential diagnosis includes diverticulosis, hemorrhoids, intestine infectious, IBS, and other intestinal tumors (lymphoma, carcinoid) [23].

### 2.4. Pathological Evaluation of Colorectal Carcinogenesis

The WHO pathologic classification divides CRC into the following histologic subtypes: glandular cancers (classic adenocarcinomas, AC, the highest percentage of diagnoses), mucinous adenocarcinomas (MAC), signet cell carcinomas (SRCC), and variants including adenosquamous carcinomas, squamous cell carcinomas, medullary carcinomas, neuroendocrine, undifferentiated and others [24]. CRC is a cancer of epithelial origin, usually developing based on conventional adenomas.

In total, 70% of all CRCs stem from adenomas. Progressions from adenoma to carcinoma take more than 10 years in sporadic cancer, whereas much shorter intervals can be observed in the Lynch syndrome [25]. Adenomas are distributed relatively throughout the colon; those with a flat or depressed morphology are distributed more in the proximal colon, and pedunculated lesions are more in the distal colon. Adenomas are, by definition, dysplastic, with the majority being low-grade; they can be characterized by tubular or villous histology, with the overwhelming majority being tubular. The increasing size of adenomas is associated with villous elements and invasive cancer (invasive cancer in adenomas ≤5 mm is extremely rare). An “advanced” stage is defined as a lesion ≥1 cm in size or having high-grade dysplasia or villous elements [26].

Serrated colorectal lesions are the next important precancerous colorectal lesions and account for up to 30% of CRCs. Currently, hyperplastic polyps (small lesions distributed toward the distal colon) are not considered precancerous, whereas sessile-serrated polyps (common, typically flat or sessile, without surface blood vessels, distributed toward the proximal colon) and traditional-serrated adenomas (rare dysplastic sessile lesion, found often in the left colon) are recognized as precancerous. The presented data are available in Figure 1 [25].

Modern molecular genetics analyses various pathogenic pathways in carcinogenesis, of which those with significant importance appear to have inflammatory cytokines, infections, mutations, and breaking out of immune surveillance. There are also confirmed by years of clinical observations of the following pre-cancerous conditions in CRC: adenomas, familial polyposis syndromes, Lynch syndrome, Crohn’s disease, and ulcerative colitis, which predisposes to the earlier development of cancer [27]. The following three main pathways in CRC carcinogenesis were proposed:(i)In 80% of cases, mutations of the APC suppressor gene are inactivated;(ii)In 15–20% of cases, there are DNA-repair gene mutations (e.g., MLH-1 promoter hypermethylation);(iii)Other mutations (e.g., TP53, BRAF mutations) [28,29,30].

Any contemporary view of molecular genetics research in the field of cancer must take into account carcinogenesis as a cellular phenomenon resulting from the interplay between genetic and epigenetic mutations. Then, their interactions with environmental factors, as well as the human microbiota influencing cellular metabolism and proliferation rate, the disturbance of genetic information, epigenetic regulation, genetic defects in chromatin remodelers, as well as limitations in knowledge about the role of non-coding regions of the genome in cancer development, should be taken together as key factors in understanding carcinogenesis [31].

### 2.5. Epigenetics of CRC

Evidence from scientific reports suggests an important role of epigenetic modifications in the development of CRC (Figure 2) [18,28,29,32,33,34,35]. Epigenetic modifications consist of changes in the methylation of cytosine-guanine (CpG) dinucleotides (DNA methylation), histone-tail post-translational modifications, and the expression of non-coding RNAs (ncRNA).

❖Chromosomal instability is known as the adenoma–carcinoma pathway. Changes include the activation of proto-oncogenes (*K-Ras*) and the inactivation of three tumor-suppression genes: the loss of *APC* (chromosome region 5q21; the most common initial gene mutated in familial/inherited and sporadic colon cancer), the loss of *p53* (chromosome region 17p13), and loss of heterozygosity for the long arm of chromosome 18 [18,28].❖The 18q Loss of Heterozygosity (LOH) is defined as the loss of one of the two copies or alleles of a gene; it is frequently found in the region of 18q21 in advanced CRC. The *DCC* (*Deleted in Colorectal Carcinoma*) gene is located on the long arm of chromosome 18 and encodes the transmembrane protein *DCC* which is called the “conditional tumor suppressor gene”. When the *DCC* gene is mutated, netrin-1 (ligand produced in the crypts of the colorectal mucosa) does not bind to the *DCC* transmembrane protein, resulting in abnormal cell survival (apoptosis disorders). *DCC* loss is related to a worse overall survival [28]. The prevalence of the allele loss of *DCC* in CRC is reported to range from 33% to 75%. LOH or a decreased *DCC* expression is associated with CRC liver and lymph node metastasis [29].❖Epigenetic Instability and CpG Methylation refer to the covalent attachment of a methyl group to the five positions of cytosine within a CpG dinucleotide in DNA. These methylation reactions are catalyzed by DNA methyltransferases (DNMTs). The regulation via inappropriate methylation of gene promoter regions is common in CRC and as significant as DNA mutations in inactivating tumor suppressor genes. The CIMP–CRC subtype (the result of tumor suppressor genes switch off, CpG island methylator phenotype) is associated with a high frequency of CpGhypermethylation, and is diagnosed based on the methylation status of various genes that participate in the regulation of calcium transport (*CACNA1G*), proliferation (*IGF2*), Wnt signaling (*NEUROG1*), transcription activity (*RUNX3*), and the suppression of cytokine signaling [28,34].❖The TP53 mutation occurs in the transition from adenoma to carcinoma. The tumor suppressor gene *p53* is mutated in almost half of all colon cancers. The *TP53* gene is involved in the control of the cell cycle and apoptosis (the loss of *p53* in colon cancers impacts the cell’s ability to both monitor centrosome integrity and regulate its duplication). Inherited or germline mutations in *TP53* are the cause of the Li-Fraumeni syndrome, a syndrome associated with a variety of neoplasms, including soft tissue sarcoma, osteosarcoma, premenopausal breast cancer, brain tumors, and adrenocortical carcinoma. There are investigations that the *TP53* gene mutation is present in node-positive CRC patients, and 5-year disease-free survival is almost double the shortness in length (35%) as *TP53*-negative patients (65%) [28,35].❖Microsatellite Instability (MSI) and Mismatch Repair (MMR) Pathways. In total, 12–17% of all CRCs have microsatellite instability, which is the system that checks and repairs the defects overlooked by DNA polymerase during DNA replication. MSI is crucial for Lynch syndrome (Hereditary Non-Polyposis Colorectal Cancer [HNPCC]) and is seen in more than 95% of cases. By contrast, for most cases of sporadic CRC, the mechanisms responsible for chromosomal instability remain elusive, and MSI is responsible for only 15–20% of the cases. MSI tumors typically occur in the proximal colon and display Crohn’s disease-like lymphocytic infiltration. MSI-H tumors are frequently poorly differentiated, but this feature is not considered to be a high-risk category. Patients with MSI-associated CRC are usually younger than 50 years old, and their survival is better than patients with other types of chromosomal alterations [28].The MMR enzymes correct errors that are missed by the proofreading function of DNA polymerase and act as an additional system to preserve genomic integrity. The loss of *MLH1* expression potentiates replication errors in microsatellite sequences. Patients with the germline loss of DNA mismatch repair capability typically develop CRC by the age of 40 in 80% of cases [28].❖Receptor X (FXR) is a nuclear transcription factor that has been recognized as a tumor suppressor protein in the intestinal mucosa. Compared to healthy colon tissue, the FXR function and expression were decreased in polyps and precancerous lesions, whereas its expression was silenced mostly at later tumor stages (I–IV) [28].❖Histone posttranslational modifications (e.g., methylation, acetylation, ubiquitination, phosphorylation) were observed in the context of CRC. The methylation of *H3K4*, *H3K36*, and *H3K79* is linked to gene expression activation, whereas *H3K9me2*, *H3K9me3*, *H3K27me3*, and *H4K20* are associated with gene repression. Wnt family member 5A (Wnt5a) was found to be downregulated in metastatic CRC [33]. The histone trimethylation at *H3K4*, *H3K9*, and *H4K20* is also associated with better prognosis in early-stage CRC. Butyrate, widely known as a histone deacetylase (HDAC) inhibitor in HT-29 colon cancer cells, prevents the activation of COX-2 transcription [33,34].❖Non-Coding RNAs refer to RNA transcripts that are not translated into proteins and are usually classified into the following two groups: (1) short ncRNA (<30 nucleotides) include microRNAs (miRNAs), siRNAs, and piwi-interacting RNAs; (2) long ncRNA (>300 nucleotides) include the long intergenic ncRNA which targets specific loci to regulate expression. The best-classified miRs negatively regulate gene expression at the posttranscriptional level (the loss of miR133a and gain of miR224 are associated with CRC tumorigenesis). Aberrant miR activity has been reported in the traditional “adenoma-carcinoma” and “serrated” models of CRC [34].❖Microbiome and CRC. Known risk factors in CRC, such as dietary habits and lifestyle, can modulate the intestine microbiome. Recent investigations suggest that the gut microbiome could be an important factor in both the prevention and etiology of CRC. The Mediterranean diet [33] was associated with changes in the host-microbiome, including increased short-chain fatty acid production and the abundance of *Prevotella* and fiber-degrading *Firmicutes*. There is a proven association between red meat consumption and the enrichment of *Bacteroidesmassiliensis*, *Alistipesfinegoldii*, and *Bilophilawadsworthia bacteria*, which may implicate CRC’s etiology [36]. The risk factor for CRC development could be a severe Salmonella infection [37]. Among the Dutch population, an increased risk of CRC in the ascending and transverse parts of the colon in patients with a reported history of Salmonella infection was observed (strongly related to infection with *Salmonella enteritidis*).

## 3. Risk Factors for Colorectal Cancer

The general risk of cancer increases with age (the median age being around 70 years) [38,39]. One of the most significant risk factors seems to be diet and its components, such as the consumption of red and processed meat, smoking, NSAIDs, diabetes type II, colitis and Crohn’s disease, and hereditary disorders (Lynch syndrome, familial adenomatous polyposis) [38,39,40].

### 3.1. Colorectal Cancer Non-Modifiable Risk Factors

Among nonmodifiable factors, the most important role for CRC’s development includes age, sex, and individual background of adenomatous polyps or inflammatory bowel disease (IBD), a familial history of CRC or adenomatous polyps, and inherited genetic risk [39,41].

The likelihood of a CRC diagnosis commonly affects males and is strongly age-related, increasing after the age of 40 and continues to rise with increasing age. More than 90% of CRC cases are diagnosed in patients aged 50 or older, and the incidence rate is more than 50 times higher in persons aged 60 to 79 years. However, CRC appears to also be increasing among younger people aged 20 to 49 years: a trend observed in high HDI economies [9].

Well-known precursor lesions of CRC are tubular and villous colorectal adenomas. The statistics show that nearly 95% of sporadic CRCs come from adenomas. Individuals who have suffered from adenomas have an increased risk of developing CRC. A long latency period estimated up to 20 years, is usually required for the development of malignancy from adenomas [42]. If adenomas are detected and removed before malignant transformation, the risk of developing CRC may be reduced.

IBD is a group of colon and small intestine inflammatory conditions; Crohn’s disease and ulcerative colitis are the principal types of IBD. The differences between ulcerative colitis and Crohn’s disease can be seen in the areas affected by those conditions. Ulcerative colitis only causes the inflammation of the mucosa of the colon and rectum, whereas Crohn’s disease affects deeper tissues causing inflammation of the full thickness of the bowel wall and may involve any part of the digestive tract from the mouth to the anus. Individuals with IBD suffer from symptoms such as pain, vomiting, and diarrhea, which worsen their quality of life. The progression of IBD leads to complications such as a toxic megacolon and bowel perforation. These conditions increase the overall risk of CRC. The relative risk of CRC has been estimated at between 4- and 20-fold but is usually caught earlier than the general population in the routine surveillance of the colon via a colonoscopy [43].

The majority of people affected by CRC do not have a family history of CRC or a predisposing illness. Nevertheless, up to 30% of those suffering from CRC have other family members who have been affected by this disease [44]. One or more first-degree relatives who have had CRC or adenomatous polyps put people at increased risk; the correlation between these is not fully understood, and scientists speculate it is a combination of genetic factors and shared environmental factors.

According to Rex et al. [25], in the CRC screening guidelines, screening is recommended to be initiated in most average-risk individuals at the age of 50 years. The recommended starting age and the frequency of staging may vary due to a family history of CRC or certain polyps. Patients with a family history of CRC in a first-degree relative diagnosed at <60 years are recommended to undergo colonoscopy every 5 years starting at age 40 years or 10 years before the relative was diagnosed; those who decline colonoscopy should be offered annual fecal immunochemical test (FIT) screening. Moreover, it is suggested that persons with one first-degree relative diagnosed with CRC or a documented advanced adenoma at age ≥ 60 years start screening at age 40. When first-degree relatives have documented advanced serrated lesions (SSPs ≥ 10 mm in size/SSP with cytologic dysplasia/traditional serrated adenoma ≥ 10 mm in size), patients should be screened according to the recommendations for people with a family history of advanced adenoma.

Population-based studies did not show rational evidence for systematic screening in asymptomatic people < 50 years old who lacked specific risk factors related to the family history of Lynch syndrome [25] (Table 2).

### 3.2. Environmental Risk Factors for Colorectal Cancer

CRC is widely considered to be an environmental disease with a wide range of ill-defined cultural, social, and lifestyle factors. CRC is one of the major cancers for which modifiable causes may be identified, and a large number of cases are theoretically prevented [6].

The research indicates that diet strongly influences the risk of CRC [33,38,41,43]. Fortunately, changes in nutritional habits might reduce up to 70% of this cancer burden. A diet that is substantial in animal fat is a major risk for CRC (with a stronger association for colon cancer than rectal cancer) [43]. Potential underlying mechanisms for a positive association between red meat consumption and CRC incidence include the presence of heme iron in red meat. Moreover, some meats are cooked at high temperatures, which results in the accumulation of heterocyclic amines and polycyclic aromatic hydrocarbons, which are both believed to have carcinogenic properties. Some studies also suggest that a diet poor in fruits, vegetables, and fiber may result in a higher risk of CRC. Epidemiological and experimental evidence highlights the preventive role of folate in carcinogenesis and shows that a higher intake of folic acid is associated with a lower risk of CRC [46,47]. Folic acid participates in DNA biosynthesis, repair, and methylation and plays an important role in cellular homeostasis [46]. It has been confirmed that a colon microbiome may influence the progression of CRC [48]. Some microbiotas mediate the effects of a specific diet on the risk of CRC by producing butyrate, folate, and biotinin, which are key in the regulation of epithelial proliferation [49] and directly modify immune activity [50]. Research shows that an imbalance of folate-producing bacteria could contribute to the development of cancer. Therefore, changes in the gut microbiome may mediate or modify the effects of environmental factors on CRC risk [46].

Physical inactivity and obesity are reported to be modifiable and interrelated risk factors that account for about a fourth to a third of CRC cases [2]. There is abundant evidence that higher levels of physical activity are associated with a lower risk of CRC. The biologic mechanisms responsible for this association are elucidated. Physical activity raises the metabolic rate and increases gut motility and maximal oxygen which increases the body’s metabolic efficiency and capacity and reduces blood pressure and insulin resistance. Being overweight or obese increases circulating estrogen decreases insulin sensitivity, and is associated with excessive abdominal adiposity.

Cigarette smoking associated with lung cancer evidence is well established [2,38,43]. Smoking is also extremely harmful to the colon and rectum—12% of CRC deaths are attributed to smoking. The carcinogens found in tobacco increase CRC and adenomatous polyps’ growth. Evidence demonstrates an earlier average age of onset incidence of CRC among men and women who smoke cigarettes.

The regular consumption of alcohol, particularly at a younger age, may be associated with an increased risk of developing CRC. Carcinogens are reactive metabolites of alcohol such as acetaldehyde; alcohol may also enhance the penetration of other carcinogenic molecules into mucosal cells. The synergistic effect of alcohol and tobacco may lead to DNA mutations. High consumption of alcohol with a low dietary intake makes their cells susceptible to carcinogenesis.

### 3.3. Inherited Genetic Risk for Colorectal Cancer

CRC is considered a complex disease, with both inherited and environmental factors involved in its predisposition. In total, 5 to 10% of CRC cases are a consequence of recognized hereditary conditions. Genome-wide association studies (GWASs) have identified over 40 genetic loci associated with CRC and adenoma risk in the general population so far. Familial adenomatous polyposis (FAP) and hereditary nonpolyposis colorectal cancer (HNPCC, LS—Lynch syndrome) are the most common inherited conditions. LS is an autosomal dominant condition, the most common cause of inherited CRC, accounting for about 3% of newly diagnosed cases of colorectal malignancy. The eponym “Lynch syndrome” recognizes Dr. Henry T. Lynch, the first author of the original 1966 publication, who comprehensively described this condition [45].

Mutations in the MLH-1 and MSH2 genes involved in the DNA repair pathway are associated with HNPCC; FAP is caused by mutations in the tumor suppressor gene APC [34,51,52]. In 1991, the International Collaborative Group on Hereditary Non-Polyposis Colon Cancer published the Amsterdam I criteria (AC-I) for defining HNPCC [45,53]. The AC-I was revised in 1996 and fulfilled if the following conditions were met:(1)Three or more relatives (one of whom is a first-degree relative) with CRC or with HNPCC-s associated cancers (endometrial carcinoma, small bowel adenocarcinoma, ureter or renal pelvis carcinoma);(2)Two successive generations affected;(3)FAP excluded;(4)Tumors confirmed in histology;(5)One or more HNPCC-related cancers diagnosed before the age of 50 years.

Half of the families that fulfill the original Amsterdam Criteria have a hereditary DNA mismatch repair gene mutation as well as the Lynch Syndrome. The other HNPCC families have no evidence of DNA mismatch repair deficiency, and studies now show that these families are different from Lynch Syndrome families. The name used to refer to the “other half of HNPCC” is Familial Colorectal Cancer Type X (FCCTX), which is undoubtedly a heterogenous grouping [45]. It likely includes some families that have a random aggregation of a common tumor; some families may be attributable to shared lifestyle factors and/or a polygenic predisposition, and some families likely have a yet-to-be-defined syndrome or an undiagnosed single-gene disorder [53]. FCCTX individuals should undergo colonoscopy every 3 to 5 years, beginning 10 years before the age at diagnosis of the youngest affected relative.

HNPCC accounts for 2–6% of CRC [43]. The lifetime risk of CRC in patients with recognized HNPCC-related mutations is higher than in the general population (70–80%), and the average age at diagnosis is lower (40–50 years) [54]. MLH-1 and MSH-2 mutations are also associated with an increased relative risk of other cancers (uterus, stomach, small bowel, pancreas, kidney, and ureter). Unlike patients with HNPCC, who develop only a few adenomas, people with FAP syndrome characteristically develop hundreds of polyps at a relatively young age; one or more of these adenomas typically undergoes a malignant transformation as early as age 20; FAP, connected with APC-associated polyposis, is inherited in an autosomal dominant manner (75–80% of individuals have an affected parent) and accounts for less than 1% of all CRC cases [55].

### 3.4. COX Role in Colorectal Cancer Carcinogenesis

In recent decades, the role of cyclooxygenase 2 (COX-2) has been appreciated in cancer development and progression. Cyclooxygenase converts arachidonic acid to prostaglandin H_2_. There are two main COX isoforms that are known: the “constitutive” isoform COX-1 and the “inducible” isoform COX-2. Since the early 1990s, there have been many publications confirming that COX-2 promotes pro-tumorigenic activity through several mechanisms: angiogenesis development and resistance to apoptosis, the modulation of host immune surveillance, increasing DNA mutagenesis, activity peroxidase activity and xenobiotic carcinogens, and promoting invasiveness.

In CRC, there is an overexpression of the COX-2 protein or mRNA compared to the surrounding normal mucosa. COX-2 overexpression is observed in up to 90% of CRCs. It is interesting that COX-2 expression is increased in adenoma and carcinoma; the COX-2 expression is higher in larger tumors and deep invasions [56,57].

The human COX-2 gene, mapped to chromosome 1q25.2-q25.3, is 8.3 kb in size, contains 10 exons, and produces an mRNA of 4.6 kb. The COX-2 gene is polymorphic, and contains a large number of single-nucleotide polymorphisms (SNPs), such as −765 G>C (rs20417), −1195 G>A (rs689466), −8473 T>C (rs5275), −1759 G>A (rs3218625), −202 C>T (rs2745557), and −1290 A>G (rs689466). COX-2 expression and COX-2 functional polymorphisms are thought to be an early event involved in colorectal cancer development.

## 4. Colorectal Cancer Treatment—A Multidisciplinary Approach

### 4.1. Colorectal Cancer Surgery

The contemporary treatment of CRC is based on the combined usage of different methods of therapy. At certain levels of disease advancement, surgical treatment, radiotherapy, and chemotherapy are administrated according to the TNM classification. There is no doubt that surgery remains the gold standard of treatment that allows the optimal goal to be achieved, which is a complete cure for the disease.

The location of the tumor in the large intestine and the severity of the neoplastic disease implies the possible method of treatment (surgery, radiotherapy, chemotherapy) (Table 3). The advances in surgical treatment we have seen in recent decades allow us to ensure that the most severe surgical complications can be avoided and the continuity of the gastrointestinal tract is maintained. The extent of intestinal resection depends on tumor localization. There are no established standards of management among patients with grade IV, and the method of surgical treatment depends on the goal that is possible to achieve. In small-group patients with the presence of peritoneal metastases as the only cancer dissemination site, hyperthermic intraperitoneal chemotherapy (HIPEC) can also be applied with a chance of success only when it is combined with the maximum cytoreductive surgical procedure (the removal of all tumor sites with a diameter above 0.5–1 cm) and in the absence of distant metastases [58,59].

Taking into consideration the stage and technical possibilities of CRC, the surgical resection procedure consists of the following [59,60]:*CRC without distant metastases—the resection of the appropriate part of the intestine together with the regional lymph nodes (en-block), depending on tumor localization and intestinal vascularization (segmental resection, hemicolectomy, dilated hemicolectomy, colectomy). Colon segment resection with the maximal length of axial mesentery vessels (containing regional lymph nodes) should be a performer in fascial anatomical spaces known as complete mesocolic excision (CME). It has to evaluate a minimum of 12 lymph nodes—all suspected lymph nodes are also removed.*CRC with the presence of synchronous resectable liver/lung metastases—the excision of the segmental intestine with the simultaneous/subsequent resection of liver/lung lesions;*CRC with the presence of synchronous unresectable liver/lung metastases (possible to resect during CTH)—the treatment starts with inductive CTH after 2–3 months of evaluating the possibility of radical surgery performance. Before the beginning of CTH, it is necessary to undergo surgery for a primary tumor if it threatens intestinal stenosis or significant bleeding;*Inoperable CRC—resection used as a palliative method (bowel resection, bypass anastomosis, debilitating colostomy, endoscopic stenosis prosthesis) [59,60,61].

The following types of resections were performed (Figure 3):

### 4.2. The Length of the Gut Resection Margins

CRC can spread an absorbent through the blood vessels through the continuity and exfoliation of tumor cells. The head principle of oncological surgeons is to not separate/split the macroscopically visible infiltration of the tumor in adjacent organs but to remove it entirely—this strategy is called “en-block resection”. However, it is of huge importance in the case of CRC infiltration that there are no cancer cells in the surgical cut line (R0 resection). It is considered that a proximal and distal intestinal margin of 5 cm in length is enough to ensure the radicality of the procedure [59,60]. This rule, however, does not apply to the distal margin length in the case of rectal cancer’s low-resection: pathomorphological evidence showed that despite the presence of tumor infiltration below the lower tumor border (DIS = distal intramural spread) states in about 2–50% of patients, only in about 5% of cases did the length of infiltration exceed 10 mm. In the case of low resection-located rectal cancer, a distal bowel margin of more than 1 cm was considered to be sufficient (under investigation) [62]. This recommendation allowed us to save sphincters and restore the gastrointestinal tract in many cases of distally located CRC. More and more clinical data prove that among selected groups of patients, this margin maybe even smaller (<1 cm) without compromising oncological radicality [62,63].

### 4.3. Lymph Node Removal in CRC Surgery

The number of lymph nodes found during surgical preparation is one of the measurable parameters of the quality of CRC surgery. There are convincing data that show that, during CRC resection, a minimum of 12 lymph nodes should be removed. However, this recommendation does not apply to rectal cancer previously under radiotherapy or preoperative chemoradiotherapy because lymph nodes may become fibrotic. In Japan, during the resection of rectal cancer, the so-called bilateral lateral lymphadenectomy is recommended (the removal of nodes located along the iliac internal and external vessels). In Europe and the United States, this is a standard unless an intraoperative examination reveals the enlargement of this group of nodes [59,60].

### 4.4. Complete Removal of Mesorectum (TME)

The implementation of the standard of CRC treatment, a new surgical technique involving the complete removal of the mesentery of the rectum (TME, *total mesorectal excision*), allowed for the reduction in the local recurrence rate after the resection of rectal cancer (from more than 30% to below 10%). Among patients with upper rectum cancer, a full cut-out of the mesorectum at the length of 5 cm below the lower tumor border (subtotal mesorectal excision) is sufficient. A complete mesenteric excision allows us to keep an optimal margin around the tumor. It is known that an excision margin length up to 1 mm is an independent unfavorable prognostic; in such cases, the postoperative pathomorphological report should contain data on both circular margin length and the macroscopic assessment of mesenteric excision according to the so-called Quirck scale [59,60,61].

### 4.5. Colorectal Cancer Systemic Treatment

Therapeutic strategies for rectal cancer have greatly progressed over the last three decades (Figure 4). Preoperative radiotherapy or neoadjuvant chemoradiotherapy (CRT) followed by complete tumor resection (total mesorectal excision—TME) is a standard treatment leading to a reduction in the local recurrence rates in locally advanced rectal cancer. Radiotherapy followed by TME is recommended in intermediate cases (cT2, cT3 without threatened factors, some cT4a) [61,64,65]. In locally advanced cases, and less often in unresectable cases, preoperative chemoradiotherapy followed by radical surgery 6–8 weeks later should be administrated. In Europe, preoperative radiotherapy is a preferred treatment option for locally advanced rectal cancer. The preoperative therapy should be given in one of the following two ways:(i)Short-course radiotherapy: 25 Gy, 5 Gy/fraction FOR 1 week followed by immediate surgery (TME < 10 days from the first radiation fraction); Europe.(ii)Long-course chemoradiotherapy: 45–50 Gy, 1.8–2 Gy/fraction without or with 5-Fluorouracil (5-FU; bolus injections with leucovorin at 6–10 times during the radiation or continuous infusion or oral capecitabine), followed by radical surgery 6–8 weeks later; United States, Canada.

The main problem is the choice of the best standard of preoperative treatment. The clinical objective is to search for predictors to identify patients best suited for preoperative treatment. Nowadays, the research is focused on the identification of molecular differences between pretreatment tumor biopsies of responders and “non-responders” to treatment.

## 5. Contemporary Diagnostics and Screening Methods—Guidelines for Colorectal Cancer

Early-stage CRCs and precancerous lesions in asymptomatic people with no prior history of cancer or precancerous lesions are usually detected via colorectal screening. Non-advanced adenomas have a very low prevalence of cancer, and their transformation from adenoma to cancer is a multi-stage process; therefore, a screening colonoscopy repeated every standardized period remains extremely useful. Large population reports recommend performing the first colonoscopy at the age of 50 years in asymptomatic patients. Currently, colonoscopy is recommended every 10 years in people with no family history of cancer. Additionally, performing an annual fecal immunochemical test should be the first screening method [25].

In addition to a physical examination, including a per-rectum examination, each patient who qualified for a colonoscopy should be provided with full information about the family history of colorectal cancer, colorectal polyps, or other cancers in the family that may indicate a genetic/familial predisposition to cancer. It should be underlined that the detection and evaluation of a primary colon tumor is only part of the examination; a total colonoscopy, along with the viewing/evaluation of the last part of the small intestine, is mandatory to detect any synchronous changes. Additional useful tests include virtual colonoscopy and CT colonography which are extremely helpful in precise tumor localization. The above-mentioned tests are indicated in patients qualified for surgical treatment with minimally invasive techniques (laparoscopic or robotic surgeries) [66].

### 5.1. Screening for Colorectal Cancer

#### 5.1.1. Colonoscopy

Colonoscopy is the most well-known and popular screening technique; it has an advantage over other screening methods because of its ability to detect and, at the same time, remove lesions suspected of a neoplastic process [23]. It is characterized by very high sensitivity in detecting any visible changes in the large intestine. This applies to both cancer and precancerous lesions. It is worth emphasizing that if no pathological changes are found during the examination, another examination may be performed after 10 years (long intervals). Reports from the USA and Germany highlight the impact of colonoscopy on a reduction in CRC incidence and mortality: 80% in the distal colon and 60% in the proximal colon [67].

Colonoscopy, like any other medical procedure, carries the risk of complications, although this risk is small. The most common complication is perforation, which occurs relatively rarely with a frequency of less than 1/1000 [68]. It should be mentioned that a well-performed colonoscopy requires proper full bowel preparation and, very often, sedation. Indisputably, multiple case–control and prospective cohort studies have shown cancer mortality to be 68% to 88% lower among people who undergo colonoscopy screening compared to those who do not [25,68].

#### 5.1.2. Fecal Immunochemical Tests (FITs)

Fecal immunochemical tests (FITs) for hemoglobin (Hb) are increasingly recommended for colorectal cancer (CRC) screening. An estimated 1–5% of large, tested populations have a positive fecal occult blood test. Of those, about 2–10% have cancer, while 20–30% have adenomas. The advantages of FIT include its non-invasive nature (easy to deliver and affordable), high sensitivity for cancer (79% in 1 meta-analysis), and low costs. The disadvantage of FIT is it needs to be repeated with poor or no sensitivity for serrated class precursor lesions [15,25].

#### 5.1.3. FIT-Fecal DNA Test

The test approved by the FDA (The U.S. Food and Drug Administration) for CRC screening is a combination of FIT and markers for abnormal DNA (aberrantly methylated promoter, regions, mutant K-ras, and β-actin). FIT-DNA has demonstrated a higher sensitivity than FIT for advanced adenomas (42% vs. 23%) and CRC (92% vs. 72%) [67].

#### 5.1.4. CT Colonography (Virtual Colonoscopy)

CT colonography (CTC) is a tool to evaluate the bowel for CRC for initial bowel screening or after FIT. It requires bowel cleansing preparation. Carbon dioxide is insufflated into the bowel using a small rectal catheter. The advantages of CT colonography include a lower risk of perforation compared with colonoscopy and a sensitivity from 82% to 92% for adenomas >1 cm in size (but for smaller lesions, the sensitivity of CTC drops to 50%). A meta-analysis suggested that symptomatic patients preferred colonoscopy as opposed to screening patients who demonstrated a preference for CTC [25,67].

#### 5.1.5. Flexible Sigmoidoscopy (FS)

FS screens for rectum and sigmoid adenomas use a flexible endoscope inserted into the distal colon. Reductions in the distal colon or rectosigmoid cancer incidence and/or mortality from 29% to 76% with FS have been confirmed through randomized trials. Flexible sigmoidoscopy has several advantages, including lower cost and risk compared with colonoscopy, more limited bowel preparation, and usually no sedation. Flexible sigmoidoscopy is recommended at 5-year intervals [25,67].

#### 5.1.6. Colon Capsule Colonoscopy (CCE)

CCE is approved for average-risk screening and is dedicated to imaging the proximal colon among patients with previous incomplete colonoscopies and for those who need colorectal imaging but have contraindications to sedation. The advantages of capsule colonoscopy are as follows: avoiding invasive procedures and avoiding the risks of colonoscopy. The disadvantage is that bowel preparation is more extensive than that for colonoscopy (European guidelines recommend the use of 4 L of polypethyleneglycol for preparation). Two meta-analyses of CCE show a sensitivity of 71–73% for the detection of all polyps and 68–69% for significant findings (defined as any polyp of >6 mm or more than 3 polyps) [25,67].

### 5.2. Per Rectum Examination

The oldest and the simplest method for rectum examination is easy to apply in primary care services, with 70% sensitivity for rectal cancer [68].

### 5.3. Blood Enzymes Testing

The Septin9 assay is a blood serum assay. *SEPT9* is located at chromosome 17q25.3. It is a conservative skeletal protein gene involved in cytokinesis and cytoskeletal organization. *SEPT9* is closely related to CRC carcinogenesis when the promoter region is hypermethylated. Once hypermethylated within CRC cells, the septin-9 protein is released into the bloodstream and can be detected via an assay. Sensitivities and specificities for detecting CRC have been reported between 52–73% and 84–91%, respectively. These detection rates were higher for late-stage cancers [25,69]. The method using the magnetic properties of nanomaterials seems to be promising in this context [70]. Recently, Hanoglu S.B. et al. proposed an electrochemical sensing surface based on the gene-based biomarker detection of the methylation levels of the specific CRC biomarker mSEPT9. Magnetic nanoparticles (MNPs), 5-methylcytosine (5-mC) antibody, and a hybridization system (Fc-PNA) were used as the base of this tool. After the optimization and characterization of patients’ serum and plasma samples, the sensing system was adapted onto a POC device [71].

The CEA tumor marker is an oncofetal antigen generated from endodermal epithelial tumor cells and is primarily used for the tumor detecting and monitoring response to therapy: (1) monitoring patients with CRC, (2) the rapid recognition of the recurrence or spread of CRC, (3) determining the survival time before palliative chemotherapy as an independent prognostic factor, (4) monitoring treatment during palliative chemotherapy, (5) determining the survival time of patients with lymph node metastases as an independent prognostic factor, (6) and the diagnosis of liver metastases (CEA growth > 20 ng/mL is a high probability of metastases in the liver within 3 months) [69,72].

CA-19-9 is recommended as a CRC tumor marker for tumor detecting and monitoring response to the therapy, usually with a correlation with CEA levels. Serum CA 19-9 is known to be elevated in various gastrointestinal cancers, such as pancreatic, gastric, hepatic, and biliary tract carcinomas [72,73]. Similarly to CEA, it is not specific to a particular histological type of neoplasm [68].

TPS (tissue polypeptide specific antigen)—a polypeptide chain, formed in the S and G2 phase of the cell cycle and released after mitosis, applied to the diagnostics and monitoring of chemotherapy in gastrointestinal tract tumors (mainly pancreatic and colorectal) and bronchial tumors [68]. The increased concentration of TPS (>90 U/L) occurs in 60–80% of patients with CRC and indicates tumor growth.

TAG-72 (tumor-associated glycoprotein)—a glycoprotein produced by endothelium cells, renal pelvis cells, gastric epithelium, and bile ducts. Diagnostic sensitivity in CRC is estimated as 28–67% [68].

### 5.4. Recommendations for CRC Screening

The most commonly used methods for CRC screening are a fecal occult blood test, colonoscopy, and sigmoidoscopy, but there is also information on the virtual colonoscopy, magnetic resonance imaging, rectal examination, or rectal enema with double contrast usage in the literature [25,67,68,69].

In Europe, the fecal occult blood test (FOBT) performed every year or every 2 years is the most frequently recommended screening test—confirmed as effective and useful in reducing CRC’s mortality rate [74]. A positive FOBT result (3–5% of subjects) is obligatorily verified with colonoscopy. To properly perform FOBT, 6 stool samples should be collected (2 specimens from 3 consecutive stools)—a positive result in at least 1/6 of the samples is an indication for a colonoscopy. The disadvantage of this method in comparison to endoscopic methods is undoubtedly its lower sensitivity and specificity in the detection of colorectal cancer, as well as lower sensitivity (11–56%) and zero specificity in the diagnosis of adenomas.

In Poland and many other countries (including the USA and Germany), colonoscopy is the primary screening test for CRC (Table 4). Colonoscopy allows the diagnosis of not only advanced cancers but also pre-cancerous conditions and allows their simultaneous removal. A per rectum examination is not a recommended screening test for non-symptomatic patients but plays an indispensable element in the physical examination of any patient with a suspected CRC [25].

## 6. The Role of General Practitioner and Prophylaxis in Colorectal Cancer Management

The knowledge of CRC risk factors provides opportunities for intervention and early prevention. The leading role in early prevention belongs to primary healthcare [75]. It is part of a novel approach to early prevention to start the anticancer battle in the patient’s environment and direct neighborhood. It seems the best place to introduce medicine, a healthy lifestyle approach, information about risk factors, and early symptoms in carcinogenesis is the institution of the General Practitioner (GP). Traditionally, the management of cancer is delivered by in-hospital specialists. In order to provide personalized and integrated care, increase cost-effectiveness, and meet the patient’s needs and expectations, policymakers, patients, and professionals advocate a transfer of cancer care from the hospital environment to the primary care setting. In countries where the GP is the gatekeeper in the care system (e.g., the Netherlands), the GP has a good personal relationship with the patient, with their current state of health and history of previous treatment. The patients often trust their GP in health matters. A good relationship between the GP and the patient has been confirmed in the GRIP study [76]. The nature of the GP’s work and working conditions create opportunities to improve the continuous and personalized care of the ever-growing number of cancer patients. All specialists involved in cancer treatment, as well as cancer patients and politicians involved in the healthcare system, point to the extremely significant position of GPs in the period of cancer management and during the follow-up [76]. An interesting study was the PEARL study, which looked at the role of GPs in cancer screening programs. Patients who had good contact with their GP were more likely to participate in screening programs, and if the invitation was ignored, an intervention/invitation from the GP was sufficient to come to these tests [77]. Oncological education and prevention are an indisputable field of action for GPs. The process of CRC carcinogenesis is long and a multi-stage process; therefore, there is enough time for prophylaxis and prevention. Prophylaxis requires the involvement of patients. Therefore, making the patient aware of what cancer is, how it develops, and how we can prevent it is essential. At this point, the role of the primary care doctor is pivotal.

## 7. Conclusions

CRC is a complex disease due to its extensive heterogeneity; thus, effective treatment could be enhanced by the implementation of a personalized medicine approach. Despite constantly improved diagnostic and individualized therapeutic methods, CRC remains one of the biggest problems of contemporary medicine. Knowledge of the basic risk factors, early clinical symptoms, and available screening tests, as well as the preservation of oncological alert, allow the proper targeting of the diagnostic process and, consequently, the earlier diagnosis of the disease. Undoubtedly, new research at the molecular and genetic level allows us to precisely understand the process of initiation and progression of cancerous diseases and, consequently, precise, personalized prevention and treatment.

## Figures and Tables

**Figure 1 ijms-25-02032-f001:**
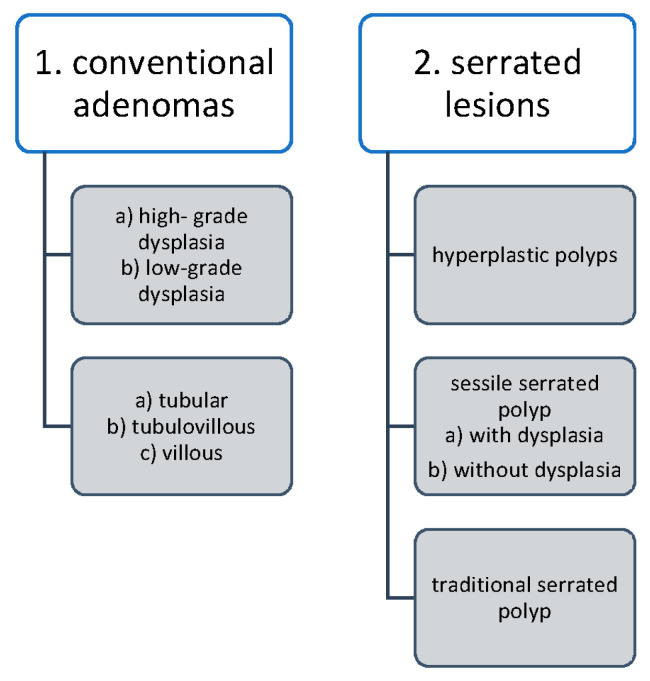
Histologic classification of major classes of colorectal polyps [25].

**Figure 2 ijms-25-02032-f002:**
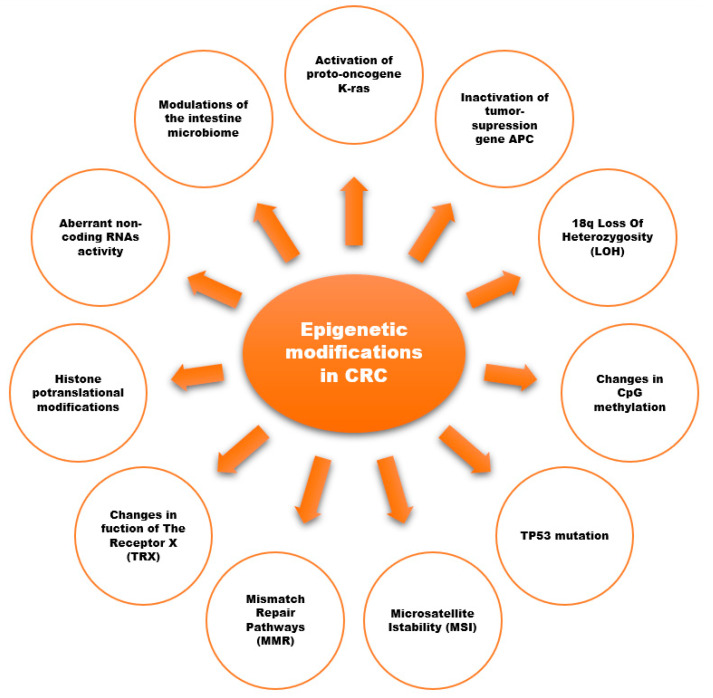
Epigenetic modifications associated with the development of CRC [18,28,29,32,33,34,35].

**Figure 3 ijms-25-02032-f003:**
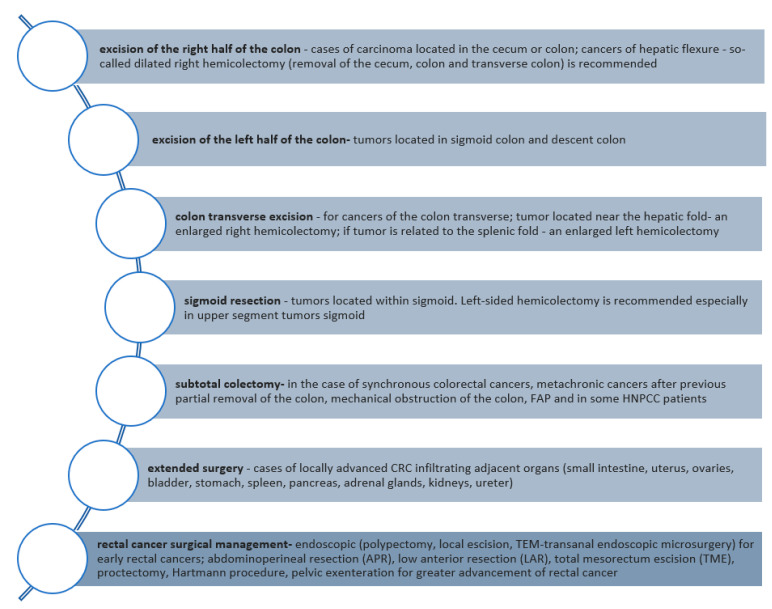
CRC treatment—types of surgical resection of the gut [59,60,61].

**Figure 4 ijms-25-02032-f004:**
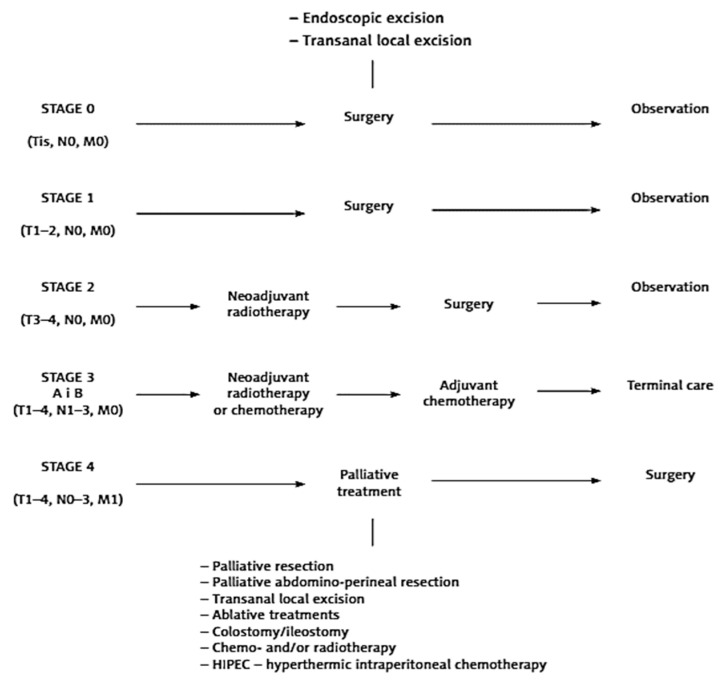
Treatment of CRC depending on the degree of clinical advancement (according to Dziki and Wallner) [61].

**Table 1 ijms-25-02032-t001:** Symptoms of CRC depending on tumor localization [18].

Right-Sided Colon Cancer	Left-Sided Colon Cancer	Rectal Cancer	Anal Cancer
non-characteristic, dull abdominal pain	flatulence, colic pain	pain in the crotch area	bleeding from the anus; itching; enlarged inguinal lymph nodes
the dark color of the stool; blood in the stool	fresh blood in the stool	fresh blood covering the stool; painful pressure on the stool	fresh blood in the stool/covering the stool; bleeding from the anal canal
iron-deficiency anemia	bowel movement disorder (constipation/diarrhea)	intestinal obstruction symptoms (vomiting, nausea)	incontinence of gases and stools
perceptible tumor	constipation; “pencil-shaped” stools	perceptible/visible tumor (per rectum examination)	perceptible/visible tumor (per rectum examination)

**Table 2 ijms-25-02032-t002:** Recommendations for people with high-risk family histories (according to the U.S Multi-Society Task Force of CRC, 2017) [25,45].

Family History	Screening Recommendations
Lynch Syndrome	Colonoscopy every 1–2 years beginning at age 20–25 years or 2–5 years younger than the youngest age at diagnosis of CRC in a family if diagnosed before age 25 years
Family Colon Cancer Syndrome X	Colonoscopy every 3–5 years beginning 10 years before the age at diagnosis of the youngest affected relative
CRC/advanced adenoma in 2 first-degree relatives diagnosed at any age OR CRC/advanced adenoma in a single first-degree relative at age < 60 years	Colonoscopy every 5 years beginning 10 years before the age at diagnosis of the youngest affected interval or age 40 (whichever is earlier)
CRC/advanced adenoma in a single first-degree relative diagnosed at age ≥ 60 years	Begin screening at the age of 40 (test and intervals as per the average-risk screening recommendations)

**Table 3 ijms-25-02032-t003:** Standards for the treatment of colorectal cancer [59,60]. CHTH—chemotherapy; RTH—radiotherapy.

Stage	Colon and Upper Part of the Rectum	Middle and Lower Part of the Rectum	Comment
Stage 0 (infiltration does not exceed the plaque muscular mucosa = high-grade dysplasia)	Endoscopy	Endoscopy	Surgical treatment in particular cases in which endoscopic treatment is not possible (too large a change, difficult location)
Stage I (cancer does not exceed full wall thickness of the intestine with lymph nodes free from cancer, without distant metastases)	Surgery	Surgery	In the case of low-located rectal cancer without infiltration sphincters can perform possible local excision from access via the anus (TEM, transanal endoscopic microsurgery)
Stage II (cancer crosses the intestinal wall and infiltrating adjacent organs; free lymph nodes with no distant metastases)	Surgery + CHTH (only in the case of bad prognostic factors (microvilli in blood vessels, lymphatic vessels or perineural vessels, low-grade tumor differentiation (G3), tumor perforation, or a low number (<12) of removed lymph nodes))	RTH or RTH/CHTH+ surgery	Surgery—the range of resection depends on the location of the tumor. Colon cancer—possible laparoscopic treatment.
Stage III (regional lymph node metastases without distant metastases)	Surgery + CHTH	RTH or RTH/CHTH+ surgery	Surgery—the range of resection depends on the location of the tumor, Colon cancer—possible laparoscopic treatment.
Stage IV (distantmetastases)	CHIR or CHTH or RTH (depending on the cancer symptoms and patient’s general condition)	CHIR or CHTH or RTH (depending on the cancer symptoms and patient’s general condition)	Palliative surgery: resection of the primary tumor or in the case of bowel obstruction (stoma or bypassing). In certain cases, cytoreductive surgery combined with the HIPEC procedure. If possible, one should strive for the radical removal of the primary tumor and metastases.

**Table 4 ijms-25-02032-t004:** Polish recommendation for CRC screening [60,72].

Polish Recommendation for CRC Screening	
Patients without CRC family history	
**Age ≥ 50 years**	Standard screening protocol
Patients with CRC family history	
**CRC in a single first-degree relative diagnosed at age ≥60 years**	Standard screening protocol started at age 40 years
**2 or more first-degree relatives diagnosed with CRC at age ≥60 OR 1 first-degree relative diagnosed at age <60 years**	Standard screening protocol started at age 40 years or 10 years before the diagnosis age of the youngest affect
**Lynch Syndrome family history**	Colonoscopy every 1–2 years beginning at age 20–25 years; colonoscopy with polyps removal every 1–2 years
**FAP**	Colonoscopy every 1–2 years

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
