# Peer review of "Managing Colorectal Cancer from Ethology to Interdisciplinary Treatment: The Gains and Challenges of Modern Medicine"

_ijms, 2024, doi:10.3390/ijms25042032_

Round 1

Reviewer 1 Report

Comments and Suggestions for Authors

The manuscript titled “Management of colorectal cancer-from ethology to interdisciplinary treatment. Gains and chellenges of modern medicine” by Berbecka, M.; et al. is a Review work where the authors to cover the state-of-the art and the most recent advances in the field of colorectal cancer diseases which is a topic of growing interest. This work could be interesting for a certain target audience and gain impact in the near future.

However, it exists some points that need to be addressed (please, see them below detailed point-by-point) to improve the scientifc quality of the submitted manuscript paper before this article will be consider for its publication in the International Journal of Molecular Sciences.

1) KEYWORDS (OPTIONAL). The authors should consider to add the term “colorectal cancer” in the keyword list.

2) Incidence of colorectal cancer and its geographic variations (lines 56-106). What are the contribution of the sex and race in this disease? Some information should be furnished in this regard.

3)  Pathological evaluation of colorectal carcinogenesis. “Modern molecular genetics analysis (…) immune surveillance” (lines 157-59). Here, the authors should point out some drawbacks linked to molecular genetic studies like the inherent interactions between genes and environment or the limited understanding of non-coding genome regions. Finally, epigenetic factors (DNA methylation, hystone modifications, ….) are already discussed in the following manuscript section.

4) “2.5. Epigenetics of CRC” (lines 171-249). A schematic representation should be added to highlight the importance and crosstalk of the epigenetic factors related to colorectal cancer.

5) “3.2. Environmental risk factors for colorectal cancer” (lines 299-397). What are the opinion of the authors about the diets rich in folate and the response of microbiome and their impact in the incidence and progres of CRC diseases? (The authors already discussed some points about the diet in the lines 304-311 albeit some further information details are missing).

6) “5.5 Colorectal cancer systemic treatment” (lines 488-513). Here, the authors listed some strategies to fight against CRC diseases. Nevertheless, it should not be neglected the role of magnetic nanoparticles [1] in the early detection of CRC diseases when they are integrated in electrochemical biosensor devices [2].

[1] Winkler, R.; et al. A Review of the Current State of Magnetic Force Microscopy to Unravel the Magnetic Properties of Nanomaterials Applied in Biological Systems and Future Directions for Quantum Technologies. Nanomaterials 2023, 13, 2585. https://doi.org/10.3390/nano13182585.

[2] Hanoglu S.B.; et al. Magnetic Nanoparticle-Based Electrochemical Sensing Platform Using Ferrocene-Labelled Peptide Nucleic Acid for the Early Diagnosis of Colorectal Cancer. Biosensors 2022, 12, 736. https://doi.org/10.3390/bios12090736.

7) CONCLUSIONS. This section clearly outlines the most relevant outcomes depicted in this Review work. However, some potential future strategic action lines should be remarked to pursue the research devoted in this topic.

Comments on the Quality of English Language

The manuscript is generally well-written albeit it may be advisable if the authors could recheck the English in order to polish final details susceptible to be improved.

Author Response

Response to the reviewer and the editor

Dear Madam/Sir,
Thank you for the favourable review and comments We did our best to improve our manuscript. Fragments of the text and bibliography have been added. All important changes are marked in green in the text. Below we enclose brief answers to yours comments:

1) KEYWORDS (OPTIONAL). The authors should consider to add the term “colorectal cancer” in the keyword list.
Answer: 
The keywords has been added.
 KEYWORDS
colorectal cancer; epidemiology; screening; prophylaxis; risk factors; biomarkers; prevention; cyclooxygenase; genes polymorphism

2) Incidence of colorectal cancer and its geographic variations (lines 56-106). What are the contribution of the sex and race in this disease? Some information should be furnished in this regard.
Answer: 
The fragment of text has been added.
… Statistics also show racial and ethnic differences in CRC survival. There was a 20% higher incident rate and approximately 40% higher mortality rate among non-Hispanic Blacks or African Americans of all racial and ethnic groups in the US. Data from the National Centers for Disease Control and Prevention show that 1 in 41 African American males will die from CRC compared to 1 in 48 for whites, and for African American women, 1 in 44 will die from CRC compared to 1 in 53 white females [7]. Research has confirmed that African Americans have a higher risk of developing CRC and a higher risk of dying from the disease [7,8]. Asian American and Pacific Islander patients has an 8% to 10% lower cancer-related mortality rate compared with non-Hispanic White patients. Among subgroups of Asian Americans and Pacific Islanders (Chinese, Japanese and Filipino), Chinese men had the lowest cancer-specific mortality [8]…..

3)  Pathological evaluation of colorectal carcinogenesis. “Modern molecular genetics analysis (…) immune surveillance” (lines 157-59). Here, the authors should point out some drawbacks linked to molecular genetic studies like the inherent interactions between genes and environment or the limited understanding of non-coding genome regions. Finally, epigenetic factors (DNA methylation, hystone modifications, ….) are already discussed in the following manuscript section.
Answer: 
The fragment of text has been added.
….Any contemporary view of molecular genetics research in the field of cancer must take into account carcinogenesis as a cellular phenomenon resulting from the interplay between genetic and epigenetic mutations and their interactions with environmental factors, as well as the human microbiota influencing cellular metabolism and proliferation rate. Disturbance of genetic information, epigenetic regulation, genetic defects in chromatin remodelers, as well as limitations in knowledge about the role of non-coding regions of the genome in cancer development should be taken together as key factors in understanding carcinogenesis [31]…..

4) “2.5. Epigenetics of CRC” (lines 171-249). A schematic representation should be added to highlight the importance and crosstalk of the epigenetic factors related to colorectal cancer.
Answer: 
Figure 2 has been added to the text.
….Figure 2. Epigenetic modifications associated with development of CRC [18,28,29,32,33,34,35]…..
5) “3.2. Environmental risk factors for colorectal cancer” (lines 299-397). What are the opinion of the authors about the diets rich in folate and the response of microbiome and their impact in the incidence and progres of CRC diseases? (The authors already discussed some points about the diet in the lines 304-311 albeit some further information details are missing).
Answer:
The fragment of text has been added. 
….Epidemiological and experimental evidence highlights the preventive role of folate in carciongenesis and shows that higher intake of folic acid is associated with a lower risk of CRC [46,47]. Folic acid participates in DNA biosynthesis, repair and methylation, and plays an important role in cellular homeostasis [46]. It has been confirmed that colon microbiome may influence the progression of CRC [48]. Some microbiota mediate the effects of a specyfic diet on the risk of CRC by producing butyrate, folate and biotinin, which are key in the regulation of epithelial proliferation [49] and directly modify immune activity [50]. Research shows that an imbalance of folate-producing bacteria could contribute to the development of cancer. Therefore, changes in the gut microbiome may mediate or modify the effects of environmental factors on CRC risk [46]…..
6) “5.5 Colorectal cancer systemic treatment” (lines 488-513). Here, the authors listed some strategies to fight against CRC diseases. Nevertheless, it should not be neglected the role of magnetic nanoparticles [1] in the early detection of CRC diseases when they are integrated in electrochemical biosensor devices [2].

[1] Winkler, R.; et al. A Review of the Current State of Magnetic Force Microscopy to Unravel the Magnetic Properties of Nanomaterials Applied in Biological Systems and Future Directions for Quantum Technologies. Nanomaterials 2023, 13, 2585. https://doi.org/10.3390/nano13182585.

[2] Hanoglu S.B.; et al. Magnetic Nanoparticle-Based Electrochemical Sensing Platform Using Ferrocene-Labelled Peptide Nucleic Acid for the Early Diagnosis of Colorectal Cancer. Biosensors 2022, 12, 736. https://doi.org/10.3390/bios12090736.
Answer: 
The fragment of text has been added.
.. Method using the magnetic properties of nanomaterials seem to be promising in this context [70]. Recently, Hanoglu S.B. et al. proposed an electrochemical sensing surface based on a gene-based biomarker detection the methylation levels of the specific CRC biomarker mSEPT9. Magnetic nanoparticles (MNPs), 5-methylcytosine (5-mC) antibody, and a hybridization system (Fc-PNA) were used as a base of this tool. After optimization and characterization of patients’ serum and plasma samples, the sensing system was adapted onto a POC device [71]…
7) CONCLUSIONS. This section clearly outlines the most relevant outcomes depicted in this Review work. However, some potential future strategic action lines should be remarked to pursue the research devoted in this topic.
Answer
The fragment of text has been added.
….Undoubtedly, new research at the molecular and genetic level will allow us to precisely understand the process of initiation and progression of cancerous diseases and, consequently, precise, personalized prevention and treatment….

The manuscript is generally well-written albeit it may be advisable if the authors could recheck the English in order to polish final details susceptible to be improved.
Answer
The authors again carefully checked the English language. The final version was accepted by all authors

Author Response

Dear reviewer,

Thank you for your review. In our opinion the review refers to another manuscript, that concerns gastric cancer but not colon cancer.